# Analysis of Anxiety, Depression and Fear of Progression at 12 Months Post-Cytoreductive Surgery in the SOCQER-2 (Surgery in Ovarian Cancer—Quality of Life Evaluation Research) Prospective, International, Multicentre Study

**DOI:** 10.3390/cancers16010075

**Published:** 2023-12-22

**Authors:** Aarti Lakhiani, Carole Cummins, Satyam Kumar, Joanna Long, Vivek Arora, Janos Balega, Tim Broadhead, Timothy Duncan, Richard Edmondson, Christina Fotopoulou, Rosalind Glasspool, Desiree Kolomainen, Ranjit Manchanda, Orla McNally, Jo Morrison, Asima Mukhopadhyay, Raj Naik, Nick Wood, Sudha Sundar

**Affiliations:** 1Institute of Applied Health Research, University of Birmingham, Birmingham B15 2TT, UK; 2Pan-Birmingham Gynaecological Cancer Centre, City Hospital, Birmingham B18 7QH, UK; janos.balega@nhs.net; 3University Hospitals Coventry and Warwickshire NHS Trust, Coventry CV2 2DX, UK; 4School of Clinical Medicine, University of New South Wales, Sydney, NSW 2052, Australia; 5Leeds Teaching Hospitals NHS Trust, Leeds LS9 7TF, UK; tim.broadhead@nhs.net; 6Norfolk & Norwich University Hospital, Norwich NR4 7UY, UK; 7Division of Cancer Services, University of Manchester, Manchester M13 9PL, UK; 8Department of Surgery and Cancer, Faculty of Medicine, Imperial College London, London SW7 2BX, UK; 9Beatson West of Scotland Cancer Centre, NHS Greater Glasgow and Clyde and School of Cancer Sciences, University of Glasgow, Glasgow G12 8QQ, UK; 10Kings College NHS Foundation Trust, London W1W 5DT, UK; 11Wolfson Institute of Population Health, Queen Mary University of London, London EC1M 6BQ, UK; 12Department of Health Services Research, London School of Hygiene & Tropical Medicine, London WC1E 7HT, UK; 13Peter MacCallum Cancer Centre, Melbourne, VIC 3052, Australia; 14Musgrove Park Hospital, Somerset NHS Foundation Trust, Taunton TA1 5DA, UK; jo.morrison@somersetft.nhs.uk; 15Kolkata Gynecological Oncology Trials and Translational Research Group, Kolkata 700156, India; 16Northern Gynaecological Oncology Centre, Queen Elizabeth Hospital, Gateshead Health NHS Foundation Trust, Gateshead NE9 6SX, UK; 17Lancashire Teaching Hospitals NHS Foundation Trust, Preston PR2 9HT, UK; 18Institute of Cancer and Genomic Sciences, University of Birmingham, Birmingham B15 2TT, UK

**Keywords:** ovarian cancer, quality of life, anxiety, depression, fear of progression

## Abstract

**Simple Summary:**

Anxiety, depression and fear of cancer progression are common psychological challenges faced by women with ovarian cancer. It can affect a person’s well-being, treatment compliance and quality of life. In this study, we assessed how often and how severe these concerns are 12 months after surgical treatment and if there is any association with surgical, patient and tumour factors. A total of 141 patients with advanced ovarian cancer who did not have disease progression at 12 months post-surgery completed two questionnaires. We found that a significant proportion of patients undergoing surgery for ovarian cancer experience anxiety, depression and fear of progression. It was not possible to identify a group of patients who are more affected by anxiety, depression or fear of progression. It is essential for healthcare providers to be attentive to the emotional needs of all ovarian cancer patients and provide appropriate support to help them cope with these psychological concerns effectively.

**Abstract:**

Patients with ovarian cancer (OC) often experience anxiety, depression and fear of progression (FOP); however, it is unclear whether surgical complexity has a role to play. We investigated the prevalence of anxiety, depression and FOP at 12 months post-cytoreductive surgery and investigated associations with surgical complexity, patient (age, ethnicity, performance status, BMI) and tumour (stage, disease load) factors. One hundred and forty-one patients with FIGO Stage III–IV OC, who did not have disease progression at 12 months post-surgery, completed the Hospital Anxiety and Depression Scale and FOP short-form questionnaire. Patients underwent surgery with low (40.4%), intermediate (31.2%) and high (28.4%) surgical complexity scores. At 12 months post-surgery, 99 of 141 (70%) patients with advanced OC undergoing surgery experienced clinically significant anxiety, 21 of 141 (14.9%) patients experienced moderate to severe depression and 37 of 140 (26.4%) experienced dysfunctional FOP. No associations were identified between the three different surgical complexity groups with regards to anxiety, depression or FOP scores. Unsurprisingly, given the natural history of the disease, most patients with OC suffer from anxiety, depression and fear of progression after completion of first-line cancer treatment. Surgical complexity at the time of surgery is not associated with a deleterious impact on anxiety, depression or FOP for patients with OC. Patients with OC experience a profound mental health impact and should be offered mental health support throughout their cancer journey.

## 1. Introduction

Ovarian cancer (OC) is the third most common gynaecological cancer and the fifth leading cause of cancer-related mortalities in women globally [1]. The majority of women are diagnosed at an advanced stage (stage III and IV) due to the natural history of the disease and vague nature of presenting symptoms, which often mimic other, less serious conditions. The diagnosis and management of OC is often emotionally distressing, especially due to the poor prognosis associated with OC and the high risk of disease progression.

Depression and anxiety are two of the most common psychological morbidities experienced by patients with cancer [2]. Patients with OC are almost twice as likely to experience clinically significant depression and more than four times as likely to experience clinically significant anxiety as women without OC [3]. Studies have shown that patients with cancer who have clinical depression and anxiety have lower treatment compliance, poorer treatment outcomes, lower quality of life (QoL), longer hospitalisation and poorer 5-year survival rates than counterparts without depression and anxiety [3,4]. A study by Colleoni et al. found there was a 40% drop in the acceptance of chemotherapy in patients with depression. Previous studies have shown that these symptoms are present during stages of diagnosis, treatment and can frequently persist after the completion of cancer treatment [5,6]. Beesley et al. reported that many women with OC still required psychological support two years post-diagnosis [7]. In the UK, the National Institute of Health and Care Excellence (NICE) guidance recommends that all patients who are diagnosed with OC should be assessed for emotional problems, given information on how to deal with emotions (e.g., depression and anxiety) and have access to appropriate psychological support services [8,9]. Despite multiple studies demonstrating significant psychological distress in patients with OC, most centres in the UK do not routinely screen for or provide any psychological support [10].

Even with improvements in care, between 70% and 75% of women diagnosed with OC will experience a recurrence or disease progression [11,12,13]. Fear of cancer progression (FOP) or fear of cancer recurrence (FCR) is defined as the “fear that the cancer will progress or return in the same place or a different part of the body” [14,15,16]. Unsurprisingly, due to the high rates of progression/recurrence, FOP is commonly reported and is a significant concern among women diagnosed with OC. A systematic review by Ozga et al. (2015) on OC and FOP found that FOP was reported as a significant concern for both older and younger women at both early and advanced stages [11].

Previous studies have shown conflicting results regarding the association between psychological distress and factors such as age and stage of disease in patients diagnosed with OC [17]. A study by Bodurka-Bevers et al. found that a poor performance score was associated with both depression and anxiety while a younger age (≤50) was only associated with depression [18]. Liu et al. showed that young age was the only independent risk factor associated with depression and anxiety in patients diagnosed with OC [19]. Norton et al. found that younger patients, along with patients diagnosed with more advanced or recurrent OC, experienced greater levels of psychological distress [20]. Price et al. identified that high symptom burden is a significant predictor for both anxiety and depression [21].

Maximal cytoreductive surgery has recently been endorsed by NICE as standard treatment for patients with advanced OC [22]. This was following publication of the results from the prospective observational SOCQER-2 study showing no detrimental impact on QoL and improved survival in centres offering maximal-effort cytoreductive surgery [23,24]. Several studies have shown improved progression-free survival (PFS) and overall survival (OS) with maximal-effort cytoreductive surgery, but we do not know its impact on patients’ psychological morbidity [25].

The primary aim of this study is to investigate the impact of maximal-effort cytoreductive surgery on anxiety, depression and fear of progression (FOP) at 12 months post-cytoreductive surgery for OC and identify any factors associated with an increased risk of psychological distress in women with OC in the SOCQER-2 cohort study. The identification of these factors may assist us in identifying patients who are most at risk and in the development of interventions to manage anxiety, depression and FOP. To our knowledge, this is the first study to investigate the impact of maximal cytoreductive surgery on mental health in patients with advanced OC.

## 2. Methods

### 2.1. Study Design

The SOCQER-2 study was a multi-centre prospective pilot observational cohort study set up to assess QoL and survival after surgery for advanced OC. Ethical approval was obtained (UK, ref. no: 15/WM/0124; India, ref. no.EC/TMC/68/16) and patients were recruited from 12 cancer centres in the UK and one centre in India over 12 months. Patients were also recruited from one centre in Australia but were not included in the analysis, as the peritoneal carcinomatosis index scores were not available, making adjustment for disease burden not possible. The study aimed to describe any impact on short-term (6 weeks), medium-term (6, 12 months) and long-term (18 months, 24 months) QoL using validated questionnaires in patients undergoing standard or extensive surgery for suspected or confirmed Stage III/IV OC. Patients were recruited before surgery and completed QoL questionnaires at prespecified time points. Patients remained in the study until 24 months or disease progression. The study found that patients undergoing maximum cytoreductive surgery for advanced OC did not experience a significant or clinically meaningful detrimental effect on global QoL compared with those undergoing less complex surgery [23]. Patients who underwent maximum cytoreductive surgery had small to moderate detriments in EORTC QLQ-C30 physical function, role function and emotional function at 6 weeks post-surgery compared with patients undergoing less extensive surgery, but by 6–12 months post-surgery, these functions were comparable across all surgical categories [23]. In addition, the SOCQER-2 study investigated the impact on anxiety, depression and fear of progression on patients who did not experience disease progression at 12 months post-cytoreductive surgery. Here, we present the results of this investigation.

### 2.2. Eligibility

All patients recruited in the SOCQER-2 study with confirmed FIGO Stage III–IV OC who did not have disease progression at 12 months post-surgery were invited to complete a Hospital Anxiety and Depression Scale (HADS) and fear of progression (FOP) short-form questionnaire at 12 months post-cytoreductive surgery.

### 2.3. Assessment of Depression and Anxiety

Depression and anxiety were assessed at 12 months post-surgery with the HADS questionnaire. HADS was originally developed for screening physically ill patients for clinically significant emotional distress [17]. It has been validated in many languages, countries and clinical settings [26,27,28]. It consists of 14 questions organised in two scales: anxiety and depression, with a score range for each scale from 0 to 21. The total score reflects anxiety or depression symptom severity with a range of 0–7, 8–10, 11–14 and 15–21, indicating no, mild, moderate and severe symptoms, respectively [27,28,29]. Moderate and severe scores have been grouped together as, in many health services, for example, the United Kingdom’s National Health Service, these categories of patients largely share the same treatment pathway [29].

### 2.4. Assessment of Fear of Progression

Patients completed the FOP short-form questionnaire at 12 months post-surgery. The FOP short-form questionnaire is a 12-item questionnaire derived from the original version [30]. The 12 items, rated on a five-point scale ranging from never to very often, form a unidimensional scale with higher scores indicating greater levels of fear [30]. Total scores ranged from 12 to 60; a cut-off of 34 or above indicates a dysfunctional level of fear of progression [31].

### 2.5. Assessment of Quality of Life

As part of the SOCQER-2 study, patients completed the validated patient-reported outcome measure EORTC QLQ-C30 at 12 months. It comprises 30 items including five functioning scales, three symptom scales, six single items and one global scale of the QoL core questionnaire. All the scales and single-item measures range in score from 0 to 100. A lower score represents a worse quality of life.

### 2.6. Statistical Analysis

Statistical analysis was performed with Stata. χ^2^, Fisher’s exact and Kruskal–Wallis tests were used to examine associations between anxiety, depression and FOP with (1) surgical (timing of surgery (PDS/NACT), surgical complexity score (SCS), outcome of surgery), (2) patient (age, Eastern Co-operative Oncology Group Performance Status (ECOG PS), age-adjusted Charlson comorbidity index, BMI, pre-surgery haemoglobin and albumin levels) and (3) tumour (FIGO stage, level of disease, presence of upper abdominal disease, Ca125, peritoneal carcinomatosis index) factors.

## 3. Results

A total of 293 patients were recruited to the SOCQER-2 study from 12 cancer centres in the UK (*n* = 235) and one centre in India (*n* = 58). After completion of their surgical intervention and final histopathology, 247 (84%) were eligible to be included in the SOCQER-2 study. Of the 247 patients, 66 patients (27%) had disease progression at 12 months and 3 patients (1%) died. Five patients (2%) withdrew consent. Therefore, 173 patients were eligible to complete the mental health impact questionnaires at 12 months. A total of 32 out of the 173 patients (13%) were lost to follow-up, leaving a total of 141 patients (Figure 1). The patients were predominantly White (66.7%), followed by those of South Asian ancestry (29.1%).

The validated Aletti surgical complexity score (SCS) was used to define surgical complexity in the SOCQER-2 study: low (score 1–3), intermediate (score 4–7) or high (score 8+). The patient characteristics of all 141 patients by SCS are presented in Table 1 (adapted from Sundar, 2022, pp. 1127–1128 [23]). A total of 40.4% of patients had low-complexity surgery, 31.2% of patients had intermediate-complexity surgery and 28.4% had high-complexity surgery.

Pre-operatively, statistically significant differences were present in the age and ECOG performance status of patients who underwent different degrees of surgical complexity. Intermediate and high SCS had more patients younger than 65 years with fewer comorbidities. In the 67% (95) of patients who had neoadjuvant chemotherapy (NACT) prior to delayed debulking surgery (DDS), 52 (55%) patients had low SCS, 29 (30%) had intermediate SCS and 14 (15%) had high SCS surgery. Among the 33% (46) undergoing primary debulking surgery (PDS), 5 (10%) patients had low SCS, 15 (33%) had intermediate SCS and 26 (57%) had high SCS surgery (*p* < 0.000) (Table 1).

### Prevalence of Depression, Anxiety and FOP

All 141 out of 141 patients (100%) completed the HADS questionnaire at 12 months. Table 2 shows the baseline and postoperative patient characteristics of all 141 patients by level of anxiety. The scores ranged from 2 to 17 with a median score of 12 and an IQR of 10 to 14. A total of 70% of patients undergoing surgery for OC experienced moderate to severe anxiety. The majority of patients experienced anxiety, but no significant associations were found with surgical, patient and tumour factors.

Table 3 shows the baseline and postoperative patient characteristics of all 141 patients by level of depression. The scores ranged from 1 to 15 with a median score of 8 and an IQR of 8 to 10. A total of 14.9% of patients undergoing surgery for OC experienced moderate to severe depression. However, 61% experienced mild depression. Three out of four patients experienced some degree of depression, but no associations were found with surgical, patient and tumour factors. 

A total of 140 out of 141 patients (99%) completed the FOP short-form questionnaire at 12 months. Table 4 shows the baseline and postoperative patient characteristics of the 140 patients by level of dysfunctional FOP. The scores ranged from 12 to 60 with a median score of 26.5 and an IQR of 20 to 34. A total of 26.4% of patients experienced dysfunctional FOP.

We investigated age, ECOG PS, age-adjusted Charlson comorbidity index, BMI, pre-surgery haemoglobin and albumin levels, FIGO stage, level of disease, presence of upper abdominal disease, Ca125, peritoneal carcinomatosis index, timing of surgery (PDS/NACT), SCS and outcome of surgery as potential associations of levels of anxiety, depression or FOP. No associations or evidence of clinically important differences were identified with any of these factors.

Fifteen patients out of 141 (8.4%) experienced both moderate/severe anxiety and depression. Twenty-three patients out of 141 (13%) experienced moderate/severe anxiety or depression only. Only 6 out of 141 patients (3.4%) experienced either anxiety or depression. Most patients experienced some level of anxiety and depression together. No clear association between the anxiety and depression scores was observed (Table 5).

Eleven out of 140 patients (8%) experienced both moderate/severe depression and dysfunctional FOP. A total of 25 out of 140 patients (18%) did not experience either depression or dysfunctional FOP, while 30% of patients with dysfunctional FOP also experienced moderate/severe depression. The level of depression and dysfunctional FOP scores were associated, with over half of those with severe depression having dysfunctional FOP (Table 6).

Seventeen out of 140 patients (12%) experienced both moderate/severe anxiety and dysfunctional FOP. Only six out of 140 patients (4%) did not experience either anxiety or dysfunctional FOP. A total of 46% of patients with dysfunctional FOP also experienced moderate/severe anxiety. The level of anxiety and dysfunctional FOP scores were negatively associated, with less than a quarter of those with moderate/severe anxiety having dysfunctional FOP (*p* < 0.001) (Table 7).

The incidence of anxiety and dysfunctional FOP in the UK is comparable to that found in India (71% vs. 68.3%, 28.3% vs. 22%). The incidence rates by ethnic group for OC in the UK patients in our study are comparable with those found in England (2013–2017) [32]. No association was found between ethnicity and anxiety, depression or FOP in White and South Asian patients, the only groups with sufficient numbers to compare (χ^2^ test). However, our patient numbers are small, especially for the ethnic minorities, and so lack statistical power.

A total of 138 out of 141 patients (97.9%) completed both the HADS questionnaire and EORTC QLQ-C30 at 12 months with a median (IQR) EORTC QLQ-C30 score of 83.33 (66.67, 83.33). The level of anxiety and EORTC QLQ-C30 scores were positively associated, with patients with anxiety having a higher QLO-C30 score (*p* < 0.046) (Figure 2a). No clear association between the level of depression and EORTC QLQ-C30 scores was observed (*p* < 0.64), (Figure 2b). A total of 137 out of 141 patients (97.2%) completed both the FOP short-form questionnaire and EORTC QLQ-C30 at 12 months. The level of FOP and EORTC QLQ-C30 scores were negatively associated, with patients with dysfunctional FOP having a lower QLO-C30 score (*p* < 0.001) (Figure 2c).

## 4. Discussion

Our study finds that, even in a cohort of patients who are cancer-free at 12 months post-surgery, the majority of patients experience clinically significant anxiety, depression and fear of progression. We were unable to identify a group of patients, either on patient or tumour factors, that are at higher risk. Patients undergoing maximum cytoreductive surgery did not experience greater mental health impact than patients undergoing less complex surgery.

Our study demonstrates that at 12 months post-surgery, three-quarters (76%) of OC patients experienced some form of depression, with a significant proportion, 14.9%, experiencing moderate to severe depression. Most (91%) patients experience anxiety at 12 months, 70% of whom experience moderate to severe anxiety. Approximately a quarter of the patients experienced dysfunctional FOP at 12 months. It was not possible to identify a subgroup of patients who are more affected by anxiety, depression or FOP from the surgical, patient and tumour factors we investigated. Unlike other studies, we did not observe an association between younger age with depression and anxiety among patients with OC [19]. It is reassuring, in view of increasing surgical complexity in practice, that the degree of surgical complexity is not associated with increased levels of anxiety, depression or FOP.

Frangou et al. (2021) found that over 50% of OC patients had symptoms of depression post-chemotherapy which improved in 3 months without any intervention [10]. Watts et al. (2015) found that the prevalence of depression in OC patients was highest (25.3%) before starting chemotherapy and lowest after the cessation of treatment (12.7%) [3]. Our study had similar findings with 14.9% of patients experiencing moderate to severe depression at 12 months post-treatment. This could indicate that depression is a direct result of the impact of a diagnosis of cancer and its treatment and resolves over time in most people.

A systematic review and meta-analysis of patients with OC found that the prevalence of anxiety in patients with OC was lowest (19.12%) before commencing treatment and highest after the cessation of treatment (27.09%) [3]. Comparably, we found that anxiety was prevalent at a very high level at 12 months. One explanation for these results may be that patients with OC experience a marked reduction in clinical consultations and support following completion of their first-line treatment as they move into the survivorship phase of their cancer journey. This can lead to increased levels of anxiety due to feelings of isolation and a fear that their cancer may return or is progressing unobserved [33,34]. The 12-month time point post-diagnosis is when most patients have completed first-line treatment and are being less intensively monitored in hospital. This is consistent with informal feedback from conversations with charity partners (personal communication, Victoria CEO Ovacome).

We also found that around a quarter of the patients experienced dysfunctional FOP. There was a negative association between dysfunctional FOP and anxiety, with less than a quarter of patients with moderate/severe anxiety experiencing dysfunctional FOP. FOP is the most frequently reported unmet psychological need of cancer patients [35]. It is associated with psychosocial outcomes, such as hopelessness, faith/spirituality, post-traumatic stress disorder (PTSD), anxiety about death and dying and uncertainty about one’s future medical health [11]. Elevated levels of FOP that become dysfunctional require support, as it can affect a patient’s well-being, impair daily activities, treatment adherence, QoL and social functioning [35,36]. Previously, FOP was believed to be linked to anxiety, but it is now considered a separate concern for patients. Instruments designed to detect anxiety or depression show inconsistent correlation with FOP scores [10,35]. A better understanding of dysfunctional FOP can help us manage this problem effectively.

We found conflicting evidence for analysis of associations with global QoL at 12 months. Dysfunctional FOP appeared associated with lower QoL; however, no association between depression and global QoL were found. Anxiety was associated with a higher QoL. Others have found that depression and anxiety were a risk factor for poor QoL at 12 months post-diagnosis [37]. The associations of mental health with global QoL are complex and nuanced and our results may be limited by the small number of patients in the study.

Our study had several strengths. Firstly, we used the validated HADS and FOP short-form questionnaires to assess symptoms of anxiety/depression and FOP, respectively, meaning misclassification between cut-points will have had minimal impact on our results. There was minimal missing data; only one patient out of the 141 patients did not complete the FOP short-form questionnaire. Secondly, the study included data from multiple cancer centres in the UK and one centre in India. Thirdly, our data consisted of high-quality data on surgical load and surgeries performed. Fourthly, only patients with no disease progression at 12 months post-surgery were included. Lastly, this is the first study that we are aware of to investigate the impact of different surgical patterns on anxiety, depression and FOP.

However, some limitations were also present. First, we only investigated the prevalence of anxiety, depression and FOP at 12 months post-cytoreductive surgery. We did not have data relating to the patient’s history of depression and anxiety pre-diagnosis, so it is impossible to determine whether a history of depression and anxiety acted as a significant precursor of current depression and anxiety [38]. It would be interesting to examine how the emotional status of these patients changes over time. A long-term longitudinal study investigating the prevalence of anxiety, depression and FOP at several time points would be beneficial in addressing these limitations. Second, only patients with advanced-stage OC were included. Third, data on some demographics that may be of potential interest were not available including marital status and educational level.

Selection bias cannot be excluded in this study; however, systematic bias introduced by surgeons recruiting patients whom they believed would recover well after extensive surgery is unlikely, as recruitment was carried out by research nurses. Although the study had high retention rates, there is some potential for bias in an unknown direction.

Previous studies have shown that maximal cytoreductive surgery improves survival in patients with advanced ovarian cancer. The SOCQER-2 study showed that high-complexity cytoreductive surgery did not result in poorer QoL compared with intermediate- or low-complexity surgery. Our study demonstrated that the use of high-complexity surgery did not have an association with anxiety, depression or fear of progression compared with less complex surgery in patients whose disease had not progressed at 12 months post-surgery. As no associations were found with the surgical, patient and tumour factors we investigated, any intervention identified needs to be targeted towards all patients.

## 5. Conclusions

The degree of surgical complexity is not associated with a deleterious impact on anxiety, depression or FOP for patients with OC compared to those undergoing less complex surgery. The majority of patients with advanced OC experience depression and FOP, and 70% experience moderate to severe levels of anxiety, even if cancer-free at 12 months post-surgery. It is not possible to identify a subgroup of patients who are more affected by anxiety, depression and FOP. These findings suggest that patients with OC need routine assessment and support for mental health from the point of diagnosis and throughout their cancer journey, regardless of the extent of disease or surgery.

## Figures and Tables

**Figure 1 cancers-16-00075-f001:**
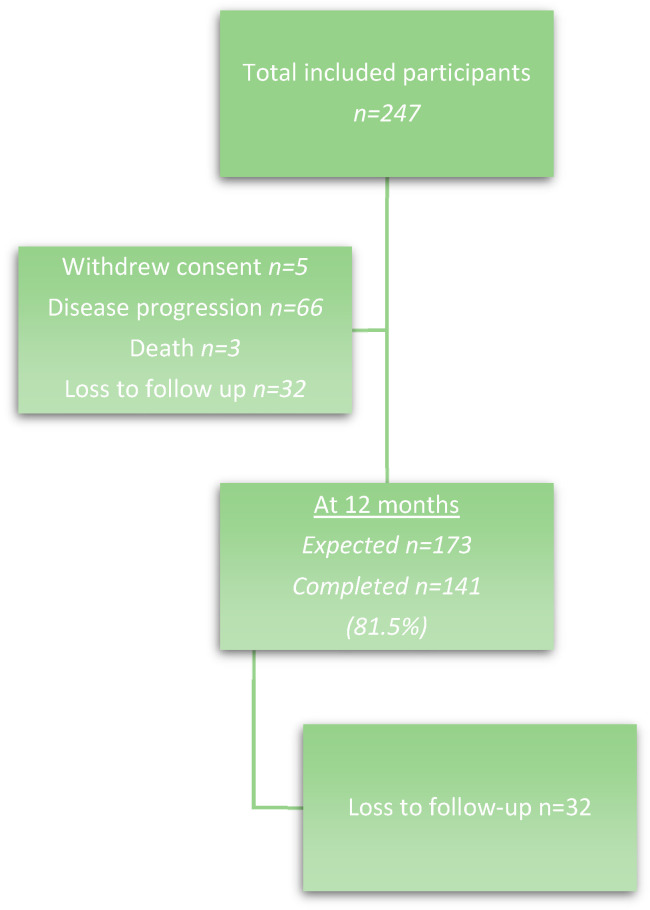
Patient flow diagram.

**Figure 2 cancers-16-00075-f002:**
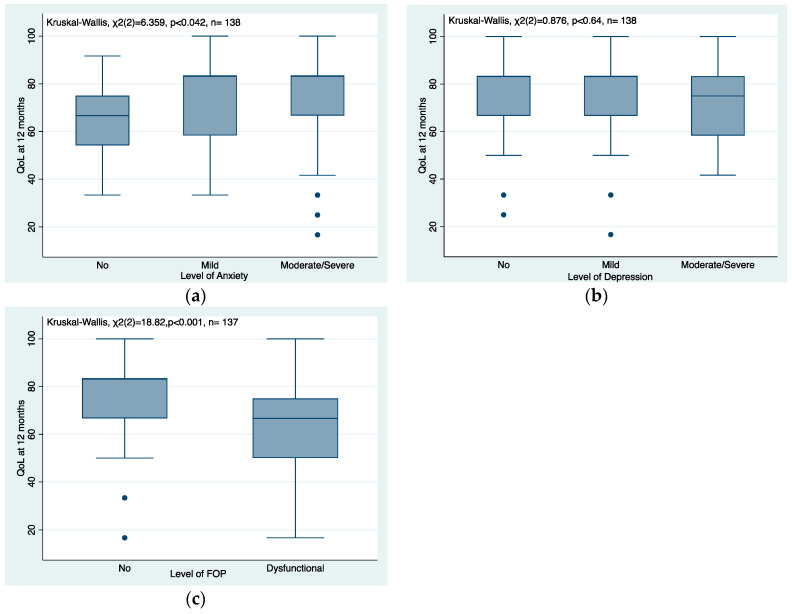
(**a**–**c**) Association between quality of life at 12 months and levels of anxiety, depression and FOP.

**Table 1 cancers-16-00075-t001:** Baseline and postoperative patient characteristics by modified Aletti surgical complexity score (SCS) group.

	Low SCS *N* = 57		Intermediate SCS *N* = 44		High SCS *N* = 40		
**Patient Characteristics**	**Number**	**%**	**Number**	**%**	**Number**	**%**	***p* Value**
**Age in years**							
**≤** **65 years**	25	43.9	27	61.4	32	80	0.002 (c)
**>** **65 years**	32	56.1	17	38.6	8	20	
**Ethnicity**							
**White**	55	96.5	26	59.1	13	32.5	<0.000 (f)
**South Asian**	2	3.5	15	34.1	24	60.7	
**Other**	0	0.0	3	6.8	3	6.8	
**Eastern Co-operative Oncology Group Performance Status**							
**0**	31	54.4	22	50.0	13	32.5	0.042 (f)
**1**	24	42.1	16	36.4	25	62.5	
**2, 3, 4**	2	3.5	6	13.6	2	5.0	
**Age-adjusted Charlson Comorbidity Index**							
**0–2**	35	61.4	29	65.9	32	80.0	0.144 (c)
**3+**	22	38.6	15	34.1	8	20.0	
**Body Mass Index (139 pts)**							
**≤** **25**	21	37.5	24	55.8	16	40.0	0.161(c)
**>** **25**	35	62.5	19	44.2	24	60.0	
**Timing of Surgery**							
**PDS**	5	8.8	15	34.1	26	65.0	<0.000 (c)
**NACT and DDS**	52	91.2	29	65.9	14	35.0	
**Pre-surgery Haemoglobin**							
**≤** **109 g/L**	23	40.3	17	38.6	14	35.0	0.866 (c)
**>** **110 g/L or above**	34	59.7	27	61.4	26	65.0	
**Pre-surgery Albumin level**							
**≤** **35 g/L**	7	12.3	6	13.6	5	12.5	0.978 (c)
**>** **35 g/L**	50	87.7	38	86.4	35	87.5	
**Baseline Ca125**							
**≤** **500**	32	56.1	15	34.1	11	27.5	0.037 (c)
**500–1000**	11	19.3	11	25.0	9	22.5	
**>** **1000**	14	24.6	18	40.9	20	50.0	
**Peritoneal Carcinomatosis Index**							
**≤** **6**	39	68.4	16	36.4	2	5.0	<0.000 (f)
**7 to 12**	9	15.8	18	40.9	3	7.5	
**>** **12**	9	15.8	10	22.7	35	87.5	
**Level/Distribution of Disease**							
**Level 1**	11	19.3	6	13.6	0	0.0	<0.000 (f)
**Level 2**	27	47.4	14	31.8	3	7.5	
**Level 3**	19	33.4	24	54.6	37	92.5	
**Presence of Upper Abdominal Disease**							
**Not Present**	37	64.9	20	45.5	3	7.5	<0.000 (c)
**Present**	20	35.1	24	54.6	37	92.5	
**Final FIGO Stage**							
**3**	42	73.7	27	61.4	22	55.0	0.083 (f)
**4**	13	22.8	16	36.3	18	45.0	
**Not Available**	2	3.5	1	2.3	0		
**Outcome of Surgery**							
**No Visible Residual Disease**	36	63.2	31	70.5	25	62.5	0.342 (c)
**Visible Residual Disease**	21	36.8	13	29.5	15	37.5	

c: χ^2^ test, f: Fisher’s exact test.

**Table 2 cancers-16-00075-t002:** Baseline and postoperative patient characteristics by level of anxiety.

	No Anxiety *N* = 13		Mild Anxiety *N* = 29		Moderate/Severe Anxiety *N* = 99		
**Patient Characteristics**	**Number**	**%**	**Number**	**%**	**Number**	**%**	***p* Value**
**Age in years**							
**≤** **65 years**	10	76.9	17	58.6	57	57.6	0.407 (c)
**>** **65 years**	3	23.1	12	41.4	42	42.4	
**Ethnicity**							
**White**	6	46.2	23	79.3	65	65.7	0.920 (f)
**South Asian**	7	53.8	6	20.7	28	28.3	
**Other**	0	0	0	0	6	6.0	
**Eastern Co-operative Oncology Group Performance Status**							
**0**	6	46.2	14	48.3	46	46.5	0.968 (k)
**1**	7	53.8	13	44.8	45	45.5	
**2, 3, 4**	0	0	2	6.9	8	8.1	
**Age-adjusted Charlson Comorbidity Index**							
**0–2**	11	84.6	23	79.3	62	62.6	0.097 (c)
**3+**	2	15.4	6	20.7	37	37.4	
**Body Mass Index (139 pts)**							
**≤** **25**	5	38.5	13	44.8	43	44.8	0.917 (c)
**>** **25**	8	61.5	16	55.2	54	54.5	
**Timing of Surgery**							
**PDS**	3	23.0	8	27.6	35	35.4	0.546 (c)
**NACT**	10	77.0	21	72.4	64	64.6	
**Pre-surgery Haemoglobin**							
**≤** **109 g/L**	6	46.2	12	41.4	36	36.4	0.736 (c)
**>** **110 g/L or above**	7	53.8	17	58.6	63	63.6	
**Pre-surgery Albumin level**							
**≤** **35 g/L**	0	0.0	6	20.7	12	13.5	0.184 (c)
**>** **35 g/L**	13	100	23	79.3	87	86.5	
**Baseline Ca125**							
**≤** **500**	8	61.5	11	37.9	39	39.4	0.236 (f)
**500–1000**	1	7.7	10	34.5	20	20.2	
**>** **1000**	4	30.8	8	27.6	40	40.4	
**Peritoneal Carcinomatosis Index**							
**≤** **6**	6	46.2	10	34.5	41	41.4	0.841 (k)
**7 to 12**	2	15.4	7	24	21	21.2	
**>** **12**	5	38.4	12	41.5	37	37.4	
**Level/Distribution of Disease**							
**Level 1**	0	0.0	3	17.6	14	14.1	0.777 (k)
**Level 2**	5	38.5	9	20.5	30	30.3	
**Level 3**	8	61.5	17	21.2	55	55.6	
**Presence of Upper Abdominal Disease**							
**Not Present**	5	38.5	12	41.5	43	43.4	0.934 (c)
**Present**	8	61.5	17	58.5	56	56.6	
**Final FIGO Stage**							
**3**	8	61.5	18	62.0	65	65.7	0.510 (f)
**4**	5	38.5	10	34.5	32	32.3	
**Not Available**	0	0.0	1	3.5	2	2.0	
**Outcome of Surgery**							
**No Visible Residual Disease**	5	38.5	21	72.4	66	66.7	0.251 (c)
**Visible Residual Disease**	8	61.5	8	27.6	33	33.3	
**Surgical Complexity Score (SCS)**							
**Low SCS**	5	38.5	12	41.5	40	40.4	0.735 (k)
**Intermediate SCS**	2	15.4	9	31.0	33	33.3	
**High SCS**	6	46.1	8	27.5	26	26.3	

c: χ^2^ test, f: Fisher’s exact test, k: Kruskal–Wallis test.

**Table 3 cancers-16-00075-t003:** Baseline and postoperative patient characteristics by level of depression.

	No Depression *N* = 34		Mild Depression *N* = 86		Moderate/Severe Depression *N* = 21		
**Patient Characteristics**	**Number**	**%**	**Number**	**%**	**Number**	**%**	***p* Value**
**Age in years**							
**≤** **65 years**	18	52.9	50	58.1	16	76.2	0.212 (c)
**>** **65 years**	16	47.1	36	41.9	5	23.8	
**Ethnicity**							
**White**	22	64.7	64	74.4	8	38.1	0.002 (f)
**South Asian**	12	35.3	17	19.8	12	57.1	
**Other**	0	0	5	5.8	1	4.8	
**Eastern Co-operative Oncology Group Performance Status**							
**0**	16	47.0	43	50.0	7	33.3	0.552 (k)
**1**	14	41.2	38	44.2	13	61.9	
**2, 3, 4**	4	11.8	5	5.8	1	4.8	
**Age-adjusted Charlson Comorbidity Index**							
**0–2**	11	84.6	23	79.3	62	64.6	0.097 (c)
**3+**	2	15.4	6	20.7	37	82.2	
**Body Mass Index (139 pts)**							
**≤** **25**	14	41.2	40	47.6	7	33.3	0.466 (c)
**>** **25**	20	58.8	44	52.4	14	66.7	
**Timing of Surgery**							
**PDS**	14	41.2	25	29.1	7	33.3	0.443 (c)
**NACT**	20	58.8	61	70.9	14	66.7	
**Pre-surgery Haemoglobin**							
**≤** **109 g/L**	13	38.2	30	34.9	11	52.4	0.335 (c)
**>** **110 g/L or above**	21	61.8	56	65.1	10	47.6	
**Pre-surgery Albumin level**							
**≤** **35 g/L**	2	5.9	15	17.4	1	4.8	0.166 (f)
**>** **35 g/L**	32	94.1	71	82.6	20	95.2	
**Baseline Ca125**							
**≤** **500**	16	47.1	32	37.2	10	47.6	0.134 (c)
**500–1000**	10	29.4	20	23.3	1	4.8	
**>** **1000**	8	23.5	34	39.5	10	47.6	
**Peritoneal Carcinomatosis Index**							
**≤** **6**	13	38.2	36	41.9	8	38.1	0.977 (k)
**7 to 12**	8	23.6	17	19.8	5	23.8	
**>** **12**	13	38.2	33	38.3	8	38.1	
**Level/Distribution of Disease**							
**Level 1**	5	14.7	10	11.6	2	9.6	0.874 (k)
**Level 2**	11	32.4	26	30.2	7	33.3	
**Level 3**	18	52.9	50	58.2	12	57.1	
**Presence of Upper Abdominal Disease**							
**Not Present**	16	47.1	35	40.7	9	42.9	0.817 (c)
**Present**	18	52.9	51	59.3	12	57.1	
**Final FIGO Stage**							
**3**	23	67.6	59	68.6	9	42.9	0.469 (f)
**4**	9	26.5	26	30.2	12	57.1	
**Not Available**	2	5.9	1	1.2	0	0.0	
**Outcome of Surgery**							
**No Visible Residual Disease**	22	64.7	55	64.0	15	71.4	0.728 (c)
**Visible Residual Disease**	12	35.3	31	36.0	6	28.6	
**Surgical Complexity Score (SCS)**							
**Low SCS**	12	35.3	39	45.3	6	28.6	0.472 (k)
**Intermediate SCS**	16	47.1	21	24.4	7	33.3	
**High SCS**	6	17.6	26	30.3	8	38.1	

c: χ^2^ test, f: Fisher’s exact test, k: Kruskal–Wallis test.

**Table 4 cancers-16-00075-t004:** Baseline and postoperative patient characteristics by level of dysfunctional FOP.

	Not Dysfunctional FOP *N* = 103		Dysfunctional FOP *N* = 37		
**Patient Characteristics**	**Number**	**%**	**Number**	**%**	***p* Value**
**Age in years**					
**≤** **65 years**	58	56.3	26	70.3	0.137 (c)
**>** **65 years**	45	43.7	11	29.7	
**Ethnicity**					
**White**	67	65.0	26	70.3	0.528 (f)
**South Asian**	32	31.1	9	24.3	
**Other**	4	3.9	4	5.4	
**Eastern Co-operative Oncology Group Performance Status**					
**0**	47	45.6	19	51.4	0.347 (k)
**1**	46	44.7	18	48.6	
**2, 3, 4**	10	9.7	0	0.0	
**Age-adjusted Charlson Comorbidity Index**					
**0–2**	66	64.1	30	81.1	0.056 (c)
**3+**	37	35.9	7	18.9	
**Body Mass Index (138 pts)**					
**≤** **25**	44	43.1	16	44.4	0.892 (c)
**>** **25**	58	56.9	20	55.6	
**Timing of Surgery**					
**PDS**	32	31.1	14	37.8	0.452 (c)
**NACT**	71	68.7	23	62.2	
**Pre-surgery Haemoglobin**					
**≤** **109 g/L**	41	39.8	13	35.1	0.617 (c)
**>** **110 g/L or above**	62	60.2	24	64.9	
**Pre-surgery Albumin level**					
**≤** **35 g/L**	14	13.6	4	10.8	0.781 (f)
**>** **35 g/L**	89	86.4	33	89.2	
**Baseline Ca125**					
**≤** **500**	41	39.8	16	43.2	0.933 (c)
**500–1000**	23	22.3	8	21.6	
**>** **1000**	39	37.9	13	35.2	
**Peritoneal Carcinomatosis Index**					
**≤** **6**	41	39.8	15	40.5	0.623 (k)
**7 to 12**	20	19.4	10	27.0	
**>** **12**	42	40.8	12	32.5	
**Level/Distribution of Disease**					
**Level 1**	13	12.6	4	10.8	0.750 (k)
**Level 2**	32	31.1	11	29.7	
**Level 3**	58	56.3	22	59.5	
**Presence of Upper Abdominal Disease**					
**Not Present**	44	46.6	15	40.5	0.818 (c)
**Present**	59	53.4	22	59.5	
**Final FIGO Stage**					
**3**	66	64.0	24	64.9	0.752 (f)
**4**	34	33.0	13	35.1	
**Not Available**	3	3.0	0	0.0	
**Outcome of Surgery**					
**No Visible Residual Disease**	64	62.1	27	73.0	0.410 (c)
**Visible Residual Disease**	39	37.9	10	27.0	
**Surgical Complexity Score (SCS)**					
**Low SCS**	45	43.6	11	29.7	0.326 (k)
**Intermediate SCS**	29	28.2	15	40.6	
**High SCS**	29	28.2	11	29.7	

c: χ^2^ test, f: Fisher’s exact test, k: Kruskal–Wallis test.

**Table 5 cancers-16-00075-t005:** Levels of anxiety by levels of depression.

	No Depression		Mild Depression		Moderate/Severe Depression		
	**Number**	**%**	**Number**	**%**	**Number**	**%**	***p* Value**
**No Anxiety**	6	17.6	5	5.8	2	9.5	0.491 (k)
**Mild Anxiety**	7	20.6	18	20.9	4	19.1	
**Moderate/Severe Anxiety**	21	61.8	63	73.3	15	71.4	
**Total**	34	100.0	86	100.0	21	100.0	

k: Kruskal–Wallis test.

**Table 6 cancers-16-00075-t006:** Levels of dysfunctional FOP by levels of depression.

	No Depression		Mild Depression		Moderate/Severe Depression		
	**Number**	**%**	**Number**	**%**	**Number**	**%**	***p* Value**
**Not Dysfunctional FOP**	25	75.8	68	79.1	10	47.6	0.103 (k)
**Dysfunctional FOP**	8	24.2	18	20.9	11	52.4	
**Total**	33	100.0	86	100.0	21	100.0	

k: Kruskal–Wallis test.

**Table 7 cancers-16-00075-t007:** Levels of dysfunctional FOP by levels of anxiety.

	No Anxiety		Mild Anxiety		Moderate/Severe Anxiety		
	**Number**	**%**	**Number**	**%**	**Number**	**%**	** *p* ** **Value**
**Not Dysfunctional FOP**	6	46.2	16	55.2	81	82.7	0.000 (k)
**Dysfunctional FOP**	7	53.8	13	44.8	17	17.3	
**Total**	13	100.0	29	100.0	98	100.0	

k: Kruskal–Wallis test.

## Data Availability

The data that support the findings of this study are available on request from the corresponding author. The data are not publicly available due to privacy or ethical restrictions.

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
