# Peer review of "Analysis of Anxiety, Depression and Fear of Progression at 12 Months Post-Cytoreductive Surgery in the SOCQER-2 (Surgery in Ovarian Cancer—Quality of Life Evaluation Research) Prospective, International, Multicentre Study"

_cancers, 2023, doi:10.3390/cancers16010075_

Round 1

Reviewer 1 Report

Comments and Suggestions for Authors

This international, multicenter study (ID: cancers-2724433) presents the results of a prospective, cohort study on the impact of maximal-effort cytoreductive surgery on anxiety, depression, and fear of progression at 12 months post cytoreductive surgery for ovarian cancer and factors associated with an increased risk of psychological distress in women with ovarian cancer.

Overall, the article follows an appropriate structure.

The Introduction section provides sufficient background and includes numerous relevant references. The goal of the work is well defined.

But, some questions (in the Methods, Results, Discussion sections) require clarification:

  • In the Methods section, subsection Assessment of Depression & Anxiety, the following is stated, I quote `The total score reflects anxiety or depression symptom severity with a range of 0-7, 8-10, 11-14, and 15- 21, indicating no, mild, moderate, and severe symptoms, respectively.`.   

The following must be specified for this sentence:

1.      Provide an explanation of why such cut-points were used for the assessment of anxiety and depression symptom levels;

2.      Cite the appropriate reference that states such cut-points for the assessment of anxiety and depression symptom levels;

3.      To explain why, after this sentence in the Methods section, a different categorization is displayed in the results section for anxiety and depression symptom levels (only 3 categories).  

  • Regarding the previous comment, the following is required to provide in the Discussion section:

4.      Discuss whether this selection of cut-points for the assessment of anxiety and depression symptom levels could have an effect on the final results in this work, that is, the absence of a significant `relationschip`;

5.      Can it be said, taking the above, the following `It was not possible to identify a subgroup of patients who are more affected by anxiety, depression, or FOP from the surgical, patient, and tumor factors we investigated. Unlike other studies, we did not observe an association between younger age with depression and anxiety among patients with OC [19]. It is reassuring in view of increasing surgical complexity in practice, that the degree of surgical complexity is not associated with increased levels of anxiety, depression, or FOP.`? Explain this.

  • In the Methods section, check and correct the Statistical analyzes subsection, so that it is in line with the Results section. Namely, the data shown in Tables 5-7 and Figure 2 represent the results marked as `Relationship`, which is used (along with the term `association`) in the description of those results and in the Discussion section. It's necessary:

6.      In the subsection Statistical analysis, it is mandatory to indicate which statistical tests were used for the assessment of `Relationship` (ie `Association`) in this manuscript.    

  • In the Results section, it is necessary to transfer the following text to the Discussion section (second paragraph on the twelfth page):

`The incidence of anxiety and dysfunctional FoP in the UK is comparable to that found in India (71% vs. 68.3%, 28.3% vs. 22%). The incidence rates by ethnic group for OC in the UK patients in our study are comparable with those found in England (2013-17) [28]. No association was found between ethnicity and anxiety, depression, or FOP in white and South Asian patients, the only groups with sufficient numbers to compare (chi-square test). However, our patient numbers are small, especially for the ethnic minorities, and so lack statistical power.`.

  • Only 3-4 new references were introduced in the Discussion section, i.e. references that were not cited in the previous part of the paper. Correct this.
  • In the paragraph on Limitations of this study, state the possibilities for overcoming the mentioned limitations. It is necessary to discuss the adequacy of the choice of cut-points as a potential limitation in this work, which may have resulted in the absence of significant differences.
  • Specifically discuss the presence of potential sources of bias in this study. 

Reviewer 2 Report

Comments and Suggestions for Authors

The Authors present the results of a very interesting multicenter study investigating anxiety and fear of progression in patients undergoing cytoreductive surgery for ovarian cancer.

the methods are very sound and the quality of presentation is high. Although no surprising results are reported, the study findings may have relevance for clinicians involved in the field.

I have two comments:

- the Authors seem to state that there is no difference in the incidence of anxiety and fear of progression between the UK and India cohorts. Can the Authors provide a deeper analysis on this, going beyond ethnicity?

- the study is multicenter. I know that it is not up to me to judge Authors' decisions, but why is there only one Author from India, compared with 15 from the UK and two from Australia (and the Australia cohort was not included)?

Round 2

Reviewer 1 Report

Comments and Suggestions for Authors

Some corrections were made in the revised version of the paper, while the authors of the paper only provided certain explanations for some comments. I thank the authors.

Clarify the following questions (Major revision):

Statistical analysis:

- Harmonize the statistical tests that are listed in the Methods section and that are used in the Results section to present the results.

- The tests listed in the Methods section, that is `chi square and Fisher's exact` only show whether there is statistical significance of differences in some values.

- The terms `association' and `relationship' can be correctly used when some other tests are applied in the paper.

- No Kruscal-Wallis test specified. Explain this.

- Under each Table, indicate which tests were used to produce the results shown on them.

Results section:

- Check and correct the NUMBERS for all variables shown for Tables 5-7. 

Round 3

Reviewer 1 Report

Comments and Suggestions for Authors

I thank the authors for their answers. I have no other comments.